

# Comparisons of forecasting for hepatitis in Guangxi Province, China by using three neural networks models

Ruijing Gan, Ni Chen and Daizheng Huang

School of Preclinical Medicine, Guangxi Medical University, Nanning, Guangxi, China

## ABSTRACT

This study compares and evaluates the prediction of hepatitis in Guangxi Province, China by using back propagation neural networks based genetic algorithm (BPNN-GA), generalized regression neural networks (GRNN), and wavelet neural networks (WNN). In order to compare the results of forecasting, the data obtained from 2004 to 2013 and 2014 were used as modeling and forecasting samples, respectively. The results show that when the small data set of hepatitis has seasonal fluctuation, the prediction result by BPNN-GA will be better than the two other methods. The WNN method is suitable for predicting the large data set of hepatitis that has seasonal fluctuation and the same for the GRNN method when the data increases steadily.

## INTRODUCTION

Hepatitis, which is an inflammation of the liver caused by a virus, is categorized into five different types: hepatitis A, B, C, D, and E. All of these viruses cause short term or acute infection; however, the hepatitis B, C, and D viruses can cause a long-term infection, called chronic hepatitis, which can lead to life-threatening complications such as cirrhosis (liver scarring), liver failure, and liver cancer (http://www.who.int/). Hepatitis causes an enormous amount of human suffering, particularly in Asia, sub-Saharan Africa, parts of the Arabian Peninsula, the South Pacific, tropical South America, and Arctic North America (*Eikenberry et al., 2009*). Viral hepatitis kills 1.5 million people every year and over one-third of the world's population (more than 2 billion people) have been or are actively infected by the hepatitis B virus (HBV) (http://www.who.int/; *Gourley, Kuang & Nagy, 2008*). It has been reported that the direct costs due to hepatitis B reach around 500 MM Yuan RMB (approximately 80 MM US dollar) in China every year (*Chinese Society of Hepatology and Chinese Society of Infectious Diseases & Chinese Medical Association, 2011*). Guangxi, officially known as Guangxi Zhuang Autonomous Region (GZAR), is a Chinese autonomous region in South Central China that is located in the southern part of the country and is bordered to Vietnam in the southwest and the Gulf of Tonkin in the south (20°54′–26°24′N, 104°26′–112°04′E). It occupies an area of 236,700 km$^2$ with a population of over 47 million people in 2014. The typical year-round climate is subtropical rainy, which consists of long, hot summers and short winters. The annual mean temperature and rainfall are 16–23 °C and 1,080–2,760 mm, respectively

Corresponding author
Daizheng Huang,
daizheng-huang@qq.com

(*Zhang et al., 2013*). Guangxi Province is a high-incidence area of viral hepatitis. Hepatitis B has been in the top three infectious diseases in Guangxi Province for the past ten years. Therefore, accurate incidence forecasting of hepatitis is critical for early prevention and for better strategic planning by the government.

Prediction of incidences of hepatitis diseases has been an ongoing effort and several complex statistical models have been offered. Zhang proposed a Nash nonlinear grey Bernoulli model termed PSO-NNGBM (1,1) to forecast the incidence of hepatitis B in Xinjiang, China (*Zhang et al., 2014*). Ren proposed a combined mathematical model using an auto-regressive integrated moving average model (ARIMA) and a back propagation neural network (BPNN) to forecast the incidence of hepatitis E in Shanghai, China (*Ren et al., 2013*). Ture compared time series prediction capabilities of three artificial neural networks (ANN), algorithms (multi-layer perceptron (MLP), radial basis function (RBF), time delay neural networks (TDNN)), and an ARIMA model to hepatitis A virus (HAV) forecasting (*Ture & Kurt, 2006*). Gan used a hybrid algorithm combining grey model and back propagation artificial neural network to forecast hepatitis B in China (*Gan et al., 2015*). A mathematical model of HBV transmission was used to predict future chronic hepatitis B (CHB) prevalence in the New Zealand Tongan population with different infection control strategies in literature (*Thornley, Bullen & Roberts, 2008*). Other studies have been performed with supervised methods for predicting viruses and pathologies (*Weitschek et al., 2012*; *Weitschek, Cunial & Felici, 2015*; *Polychronopoulos et al., 2014*).

We note that nonlinear relationships may exist among the monthly incidences of hepatitis. While the ARIMA model can only extract linear relationships within the time series data and does not efficiently extract the full relationship hidden in the historical data. The ANN time series models can capture the historical information by nonlinear functions (*Zhang et al., 2013*).

An ANN employs nonlinear mathematical models to mimic the human problem-solving process by learning previously observations to build a system of "neurons" that makes new decisions, classifications, and forecasts (*Tang et al., 2013*; *Terrin et al., 2003*). The ANN model has been successfully used to predict hepatitis A (*Guan, Huang & Zhou, 2005*).

The aim of this study was to use three neural networks methods, namely, back propagation neural networks based on genetic algorithm (BPNN-GA), wavelet neural networks (WNN), and generalized regression neural networks (GRNN) to forecast hepatitis in the Guangxi Province of China, and compare the performance of these three methods. This comparison may be helpful for epidemiologists in choosing the most suitable methodology in a given situation.

## MATERIALS AND METHODS

### Materials

The incidence of hepatitis data, including hepatitis A, B, C, and E, were collected on a monthly base from the Chinese National Surveillance System (http://www.gxhfpc.gov.cn/xxgks/yqxx/yqyb/list_459_2.html) and the Guangxi Health Information Network (http://www.phsciencedata.cn/Share/ky_sjml.jsp?id=8defcfc2-b9a4-4225-b92c-ebb002321cea&show=0) from January 2004 to December 2014. These data composed the time series

$X = \{x(0), x(1), \ldots, x(131)\}$. The information belongs to the government statistical data and is available to the public. Hepatitis D has not been considered because the data cannot be obtained from the Chinese National Surveillance System and the Guangxi Health Information Network. The incidence dataset between 2004 and 2013 was used as the training sample to fit the model, and the dataset in 2014 was used as the testing sample.

## Methods

Three ANN methods: BPNN-GA, WNN, and GRNN, were used for prediction and their performances were compared.

### BPNN-GA model

BPNN is a multi-layered feed-forward neural network; the main features are that the signal transports forward, and the error transports backward. The input signal will be processed layer-by-layer from the input layer to the output layer. The next state of the neuron is only affected by the front state of the neuron in the layer. If the expected output was not received, the weights and the thresholds of the network will be adjusted by the error that transports backward. Therefore, the desired output will be achieved in an iterative manner (*Ramesh Babu & Arulmozhivarman, 2013*).

If the model of BPNN has $i$ input nodes, $j$ hidden nodes, and $k$ output nodes, there will be weight variables of dimensionality $N = i \times j$ between the input layer and the hidden layer, $j \times 1$ threshold variables in hidden layer, weight variable of dimensionality $M = j \times k$ between the hidden layer and the output layer, and $k \times 1$ threshold variables in the output layer (*Huang, Gong & Gong, 2015*). The topology structure is shown in Fig. 1.

Genetic algorithm (GA) is a search heuristic that mimics the process of natural selection. This heuristic is routinely used to generate useful solutions to optimization and search problems (*Mitchell, 1996*). The initial weights and thresholds of BPNN are optimized by GA, which is called the BPNN-GA method. The algorithm of the BPNN-GA flow chart is shown in Fig. 2.

### WNN model

WNN is a kind of neural network with a structure that is established on the basis of BPNN, and the wavelet basis function is taken as the transfer function is in hidden layer nodes. The signal also transports forward and the error transports backward. The topology structure is shown in Fig. 3. WNN includes two new variables, a scale factor, a displacement factor, which give it excellent functional approximation. The WNN method is composed of relatively less expensive terms that often has fast functional approximation abilities and good predicting precision (due to its ability to sift out the parameters). Compared to BPNN, the weight coefficient of WNN has the characteristics of linearity, and the objective function of learning has the feature of convexity. These properties will avoid being nonlinear in local optimization when the network is trained (*Antonios & Zapranis, 2014*; *Alexandridis & Zapranis, 2013*).
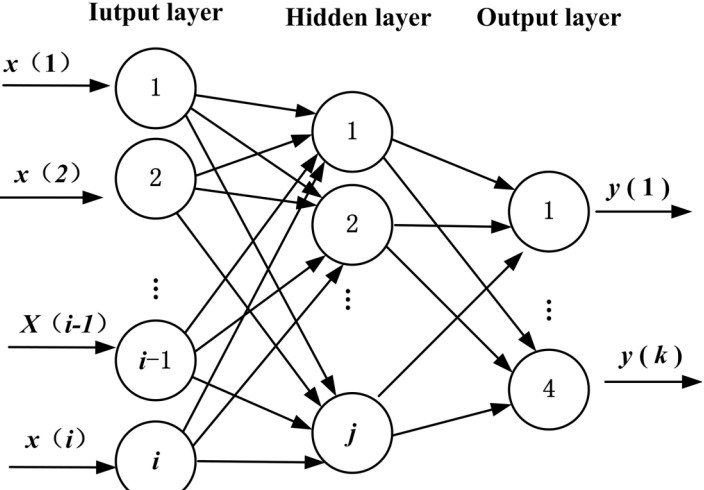

Figure 1 Topology structure of BPNN.

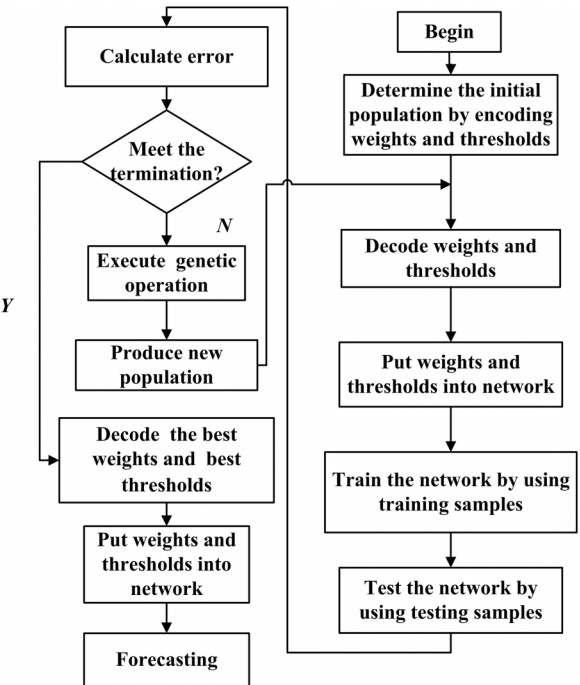

Figure 2 Flow chart of the BPNN prediction algorithm optimized by GA.

The formula for calculating the hidden layer for an input signal sequence is $x_i(i = 1, 2, \ldots, k)$ is as follows:

$$h(j) = h_j \left[ \frac{\sum_{i=1}^{k} \omega_{ij} x_i - b_j}{a_j} \right] \quad j = 1, 2, ..., l$$

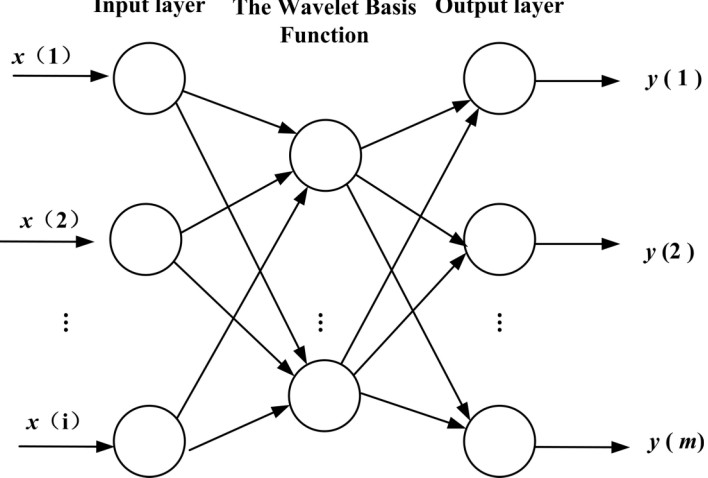

**Figure 3 Topology structure of WNN.**

where $k$ is the number of input signal; $l$ is the number of nodes in the hidden layer; $h(j)$ is the output of the $j$th node in the hidden layer; $h_j$ is the wavelet basis function; $\omega_{ij}$ is the weights between the input layer and hidden layer; $a_j$ is the scale factor of $h_j$ and $b_j$ is the displacement factor of $h_j$.

The formula to calculate the output layer is as follows.

$$y(k) = \sum_{i=1}^{l} \omega_{ik} h(i) \ k = 1, 2, ..., m$$

where $h(i)$ is the output of the $i$th node in the hidden layer; $l$ is the number of nodes in the hidden layer; $m$ is the number of nodes in the output layer; and $\omega_{ij}$ is the weight between the hidden layer and the output layer.

The weights of the network, the scale factor, and the displacement factor were estimated by the steepest descent method in WNN. The correction process of prediction used by WNN follows.

**Step 1.** Calculate error of prediction.

$$e = \sum_{k=1}^{m} yn(k) - y(k)$$

where $yn(k)$ is the expected output, namely the true value. $y(k)$ is the forecasting output.

**Step 2.** Correct the weight of the network and the coefficients of wavelet basis function according to the prediction error.

$$\omega_{n,k}^{(i+1)} = \omega_{n,k}^{i} + \Delta\omega_{n,k}^{(i+1)}$$

$$a_k^{(i+1)} = a_k^{i} + \Delta a_k^{(i+1)}$$

$$b_k^{(i+1)} = b_k^i + \Delta b_k^{(i+1)}$$

where

$$\Delta \omega_{n,k}^{(i+1)} = -\eta \frac{\partial e}{\partial \omega_{n,k}^{(i)}}$$

$$\Delta a_k^{(i+1)} = -\eta \frac{\partial e}{\partial a_k^{(i)}}$$

$$\Delta b_k^{(i+1)} = -\eta \frac{\partial e}{\partial b_k^{(i)}}$$

and $\eta$ is the learning rate.

The algorithm of WNN flow chart is shown in Fig. 4.

## GRNN model

GRNN is a memory-based network that provides estimates of continuous variables and converges to the underlying (linear or nonlinear) regression surface (*Specht, 1991*). One advantage of it is the simplicity. The adjustment of one parameter, namely, the spreading factor, is sufficient for determining the network.

The topology structure of GRNN consists of four layers: the input layer, the pattern layer, the summation layer, and the output layer. The topology structure is shown in Fig. 5.

The number of neurons in the input layer is equal to the dimension of the input vector of the learning samples. Every neuron in the input layer is the simple distribution unit and directly transmits the input variables to the pattern layer.

The neurons in the pattern layer and the neurons in the input layer have the same number and every one of the neurons in the pattern layer corresponds to a different sample. The transfer function of neurons in the pattern layer is as follows:

$$P_i = e^{-\frac{(X-X_i)^T(X-X_i)}{2\sigma^2}} \quad i = 1, 2, ..., n$$

where $P_i$ is the output of neurons in the pattern layer; $X = [x_1, x_2, \ldots, x_n]$ is the input vector; $X_i$ is the learning samples of the $i$-th neurons; $n$ is the number of input; $i$ is the number of neurons; and $\sigma$ is the smoothness factor.

There are two kinds of summation for the neurons in the summation layer. The first one is that the arithmetic sum is calculated for the output of neurons in the pattern layer. The weight between the pattern layer and every neuron is 1. The transfer function is shown in formula as follows.

$$S_D = \sum_{i=1}^{n} P_i = \sum_{i=1}^{n} e^{-\frac{(X-X_i)^T(X-X_i)}{2\sigma^2}} \quad i = 1, 2, ..., n$$

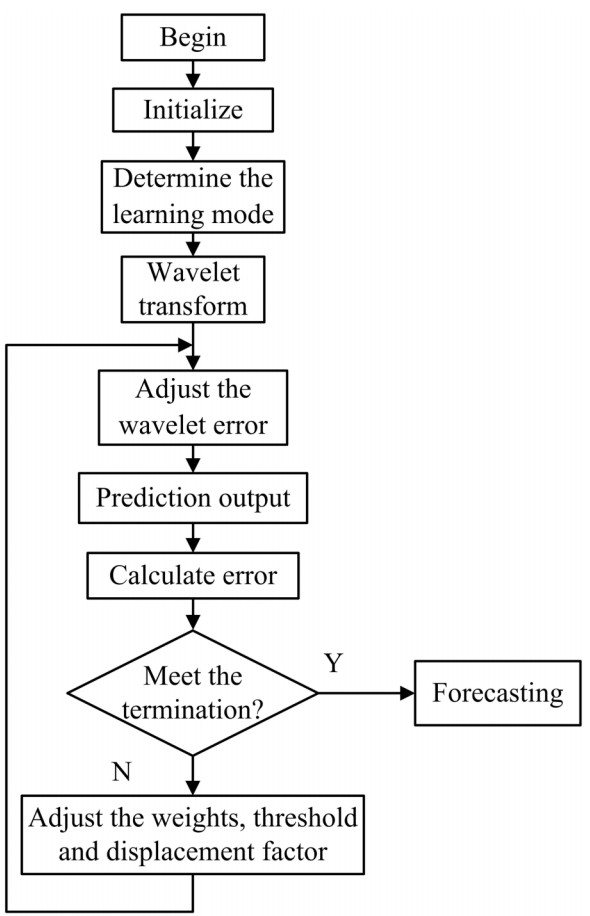

**Figure 4 Flow chart of the WNN prediction algorithm.**

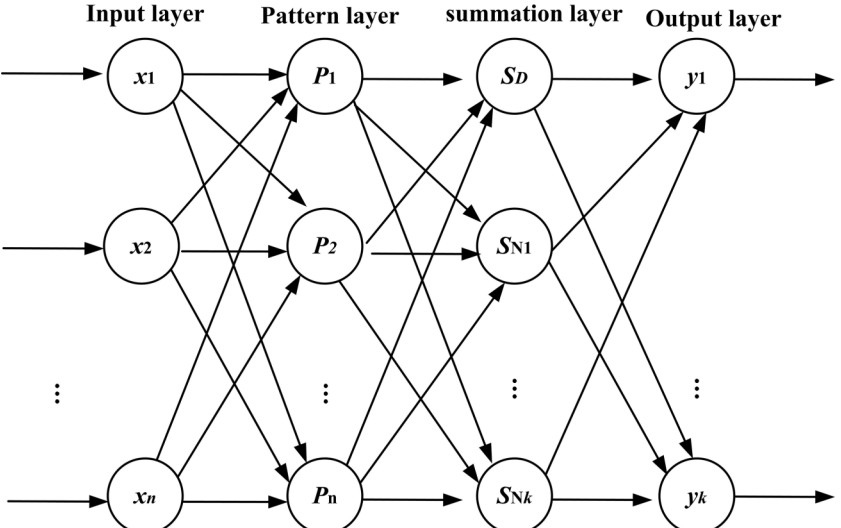

**Figure 5 Topology structure of GRNN.**

The second one is the weighted sum performed for the output of neurons in the pattern layer. The weight between the $i$-th neuron in the pattern layer and the $j$-th summation neuron is equal to the $j$-th element in the $i$-th output samples of $Y_i$. The transfer function is given below:

$$S_{Nj} = \sum_{i=1}^{n} Y_{ij}P_i = \sum_{i=1}^{n} Y_i e^{-\frac{(X-X_i)^T(X-X_i)}{2\sigma^2}} \quad j = 1, 2, ..., k$$

where $k$ is the dimension of the output vector.

The number of neurons in the output layer is equal to the dimension of the input vector of the learning samples. The output of the $j$-th neurons is shown in formula as follows.

$$y_i = \frac{S_{Nj}}{S_D} \quad j = 1, 2, ..., k$$

## RESULTS

The incidence of hepatitis that took place in Guangxi Province from January 2004 to November 2014 is considered as the original time series $X = \{x(0), x(1), . . . , x(131)\}$ and is shown in Fig. 6.

The incidence dataset between 2004 and 2013 was used as the training sample to fit the model, and the dataset in 2014 was used as the testing sample.

Of all three types of ANN, the optimal four layer neurons were experimentally selected and have average square error less than 0.01. The output layer only contains one neuron representing the forecast value of the incidence of the next month.

The hidden node $n_2$ and the input node $n_1$ in the three-layer BPNN-GA were related by $n_2 = 2n_1 + 1$ and a three-layer BPNN-GA model with four input nodes, nine hidden nodes, and one output node (4-9-1) was obtained. The selection for parameters of BPNN and GA are based on the literature *Zhang & Suganthan (2016)* and *Azadeh et al. (2007)*, respectively. S-tangent function tansig() and S-log function logsig() were used as transfer functions of the hidden layer neurons and the output layer neurons, respectively. The error between the training output and the expected output (actual output) was 0.001, learning rate was 0.9, momentum factor was 0.95, the training time was 1,000 iterations, and the parameters of GA were as shown in Table 1.

There were four input nodes, six hidden nodes, and one output node in (4-6-1) WNN. The weights of the network, the scale factor, and the displacement factor were estimated by the steepest descent method. The initial weight was 0.01, learning rate of parameter was 0.001, and the number of iterative learning was 100. The mother wavelet basis function of Morlet was used in the paper which is shown as follows.

$$y = \cos(1.75x)e^{-x^2/2}$$

Four-fold cross validation was experimentally selected and has the best prediction, which is employed to train the GRNN model and the optimal spreading factor was calculated by looping from 0.1 to 2 intervals 0.1. The transfer function of the

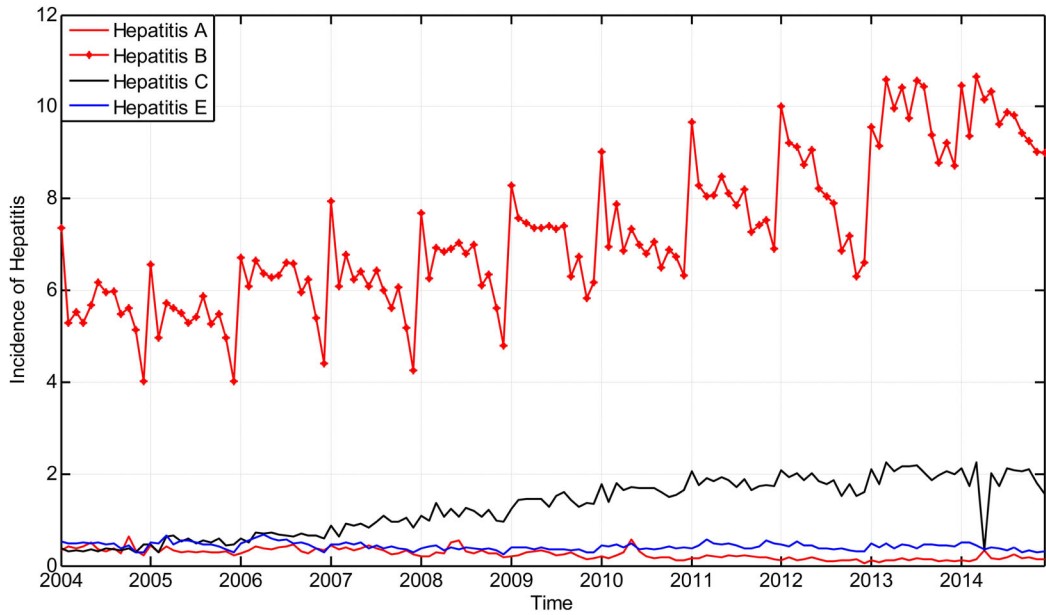

**Figure 6 The main incidence of hepatitis in Guangxi Province, China from January 2004 to December 2014.**

**Table 1 Parameters of the GA used to optimize the BPNN.**

| Population size | 40 |
| --- | --- |
| Algebra | 50 |
| Number of bits | 10 |
| Crossover probability | 0.7 |
| Mutation probability | 0.01 |
| Generation gap | 0.95 |

summation layer neurons used in the paper is shown as follows.

$$S_D = \sum_{i=1}^{n} P_i = \sum_{i=1}^{n} e^{-\frac{(X-X_i)^T(X-X_i)}{2\sigma^2}} \quad i = 1, 2, ..., n$$

where $P_i$ is the output of the pattern layer neurons; $X = [x_1, x_2, \cdots, x_n]^T$ is the input vector; $X_i$ is the learning samples of the $i$-th neurons; $n$ is the number of input; $i$ is the number of neurons; and $\sigma$ is the smoothness factor.

The contrast between the observed values and the predicted values obtained through the three methods are shown in Fig. 7.

## DISCUSSION

### The relationship between predictions and seasonal fluctuation index

The seasonal fluctuation index of incidence is used to reveal the fluctuations of incidence with seasons. The seasonal fluctuation index of the same month in eleven years from
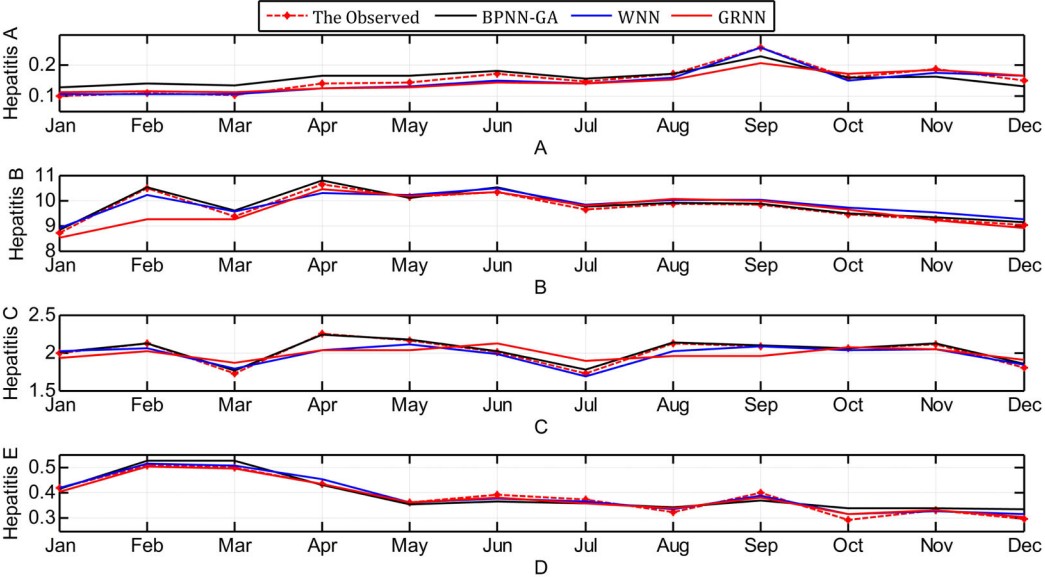

**Figure 7 Contrast between observed values and predicted values using the three methods.**

2004 to 2014 can be calculated as:

$$SFI1 = \frac{|\overline{x}_{same} - \overline{x}_{all}|}{\overline{x}_{all}}$$

where $\overline{x}_{same}$ is the average incidence of the same month and $\overline{x}_{all}$ is the average incidence of all of the months from 2004 to 2014.

The seasonal fluctuation index of the every month in 2014 is calculated as:

$$SFI2 = \frac{|x_i - \overline{x}|}{\overline{x}}, \ i = 1, ..., 12$$

where $x$ is the incidence in each month and $\overline{x}$ is the average incidences of all of the months in 2014.

Obviously, the greater the number that the seasonal fluctuation index is, the more seasonal volatility of incidence is. That is to say, the index changes reflect the disease variation in the different months. In order to compare the relationship between the seasonal fluctuation index of incidence and the three prediction results, the relative error of prediction is defined as:

$$RE_i = \frac{|\hat{y}_i - y_i|}{y_i}, \ i = 1, 2, \ldots, n$$

where $\hat{y}_i$ is the predicted value and $y_i$ are the observed values.

The seasonal fluctuation index of incidence and the relative error of the three prediction results are shown in Fig. 8.

Looking at Fig. 8, it can be seen that: 1) hepatitis A, B, and E have obvious seasonal characteristics. For Hepatitis B, in particular, the incidence which happens annually in January and February is relatively high with a rapid decline in March. April to September

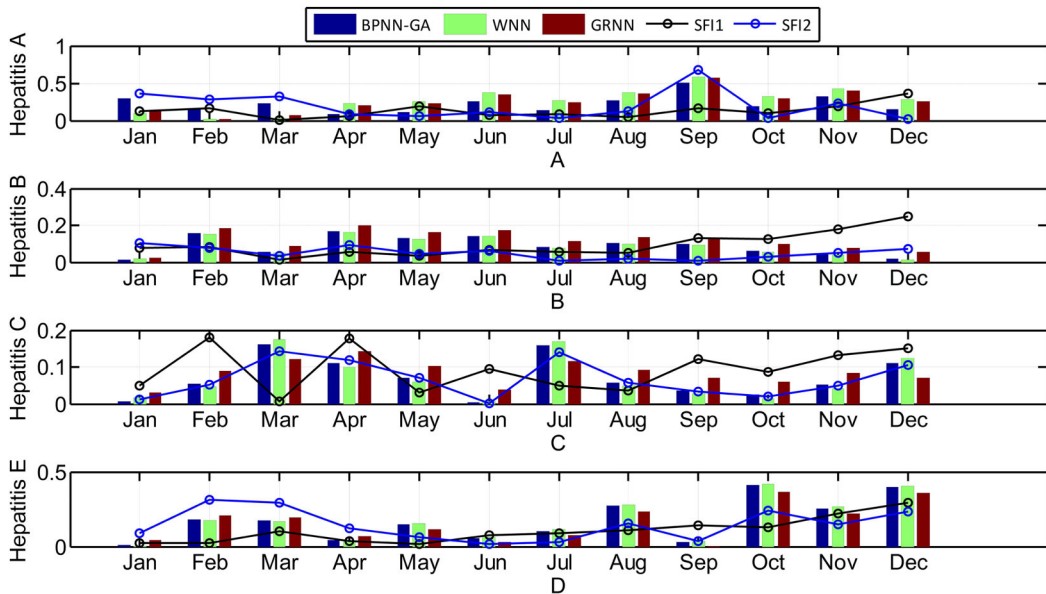

**Figure 8 The relationship between the seasonal fluctuation index and RE of the predictions by the three methods (Histograms and curves represent RE of the predictions and the seasonal fluctuation index, respectively).**

is relatively stable, but from October to December it began to rise significantly; 2) the greater the seasonal fluctuation index of the every month in 2014, the greater the relative error, especially in hepatitis C and E, which shows that the greater the disease fluctuations, the worse the prediction results; 3) the absolute error of the BPNN-GA is smaller than that of the other two methods when the incidence data is stable, such as from April to August for hepatitis A; the absolute error of the GRNN is smaller than that of the other two methods when the incidence data has great fluctuation, such as March, July, and December for hepatitis C and August, October, November, and December for in hepatitis E; and 4) the absolute error of the GRNN is larger than that of the other two methods when the incidence data is larger, and the absolute error of the WNN is larger than that of the other two methods when the incidence data is smaller. The size relationship of the average incidence is: B > C > E > A. When the incidence data is large, such as the data for hepatitis B, the size relationship of the absolute error of three methods is: GRNN > BPNN-GA > WNN.

## Comparison of evaluation indexes

The mean square error (MSE), root mean square error (RMSE), mean average error (MAE), mean average percentage error (MAPE), and sum of squared error (SSE), have been calculated and compared with the three methods. The performance indexes are defined as shown in the following.

$$MSE = \frac{1}{n}\sum_{i=1}^{n}(\hat{y}_i - y_i)^2$$

$$MAE = \frac{1}{n}\sum_{i=1}^{n}\left|\hat{y}_i - y_i\right|$$

$$RMSE = \sqrt{\frac{1}{n}\sum_{i=1}^{n}\left(\hat{y}_i - y_i\right)^2}$$

$$MAPE = \frac{1}{n}\sum_{i=1}^{n}\left|\hat{y}_i - y_i\right| * 100$$

$$SSE = \sum_{i=1}^{n}\left(\hat{y}_i - y_i\right)^2$$

where $\hat{y}_i$ is the predicted value and $y_i$ are the observed value.

The main evaluation indexes that were calculated by these three methods are listed in Table 2.

From the definitions of the other evaluation indexes, including MSE, MAE, RMSE, SSE, and MAPE, we know that the smaller the values of these indexes are, the more accurate the prediction is. The BPNN-GA method had the smallest values of these evaluation indexes when it was used to predict hepatitis A and the WNN method had the smallest values of these indexes when it was used to predict hepatitis B; the same for the GRNN method when it was used to predict hepatitis E. It can be seen that the BPNN-GA and WNN methods were not superior to the others when they were used to predict hepatitis C, but they were all superior to GRNN method. According to Fig. 8, we know that: 1) hepatitis A, B, and E have a strong seasonal volatility, but hepatitis C fluctuates up and down monthly and does not have seasonal volatility; and 2) the incidence data of hepatitis A and B are the smallest and the largest, respectively. Hepatitis E increased slowly from January to December (except for March). That is to say, these three prediction methods have their advantages when they are used to predict seasonal fluctuation data. The BPNN-GA and WNN methods are suitable for predicting small and large data, respectively, while GRNN is suitable for predicting data that increases steadily. The BPNN-GA and WNN methods were not superior to the others when they were used to predict the data that fluctuated up and down monthly and does not have seasonal volatility, and the GRNN method is not suitable for predicting these types of data.

## Comparison of statistical significance tests

Statistical significance of the obtained results was investigated using T-test; a p-value of < 0.05 was considered significant. The results are listed in Table 3.

The correlation will be better when the correlation coefficient is close to 1, namely, the predicted value is closer to the observed value. From Table 3, it can be seen that the BPNN-GA method has the best correlation when it was used to predict hepatitis B and C.

**Table 2 Comparison of the evaluation indexes in the prediction results.**

| Hepatitis | Method | MSE | MAE | RMSE | SSE | MAPE |
|---|---|---|---|---|---|---|
| A | BPNN-GA | **0.0024** | **0.0377** | **0.0488** | **0.0286** | **3.7743** |
| | WNN | 0.0038 | 0.0480 | 0.0616 | 0.0455 | 4.7955 |
| | GRNN | 0.0034 | 0.0456 | 0.0587 | 0.0413 | 4.5566 |
| B | BPNN-GA | 1.1018 | 0.9008 | 1.0497 | 13.2217 | 90.0830 |
| | WNN | **1.0285** | **0.8652** | **1.0141** | **12.3414** | **86.5163** |
| | GRNN | 1.7907 | 1.2085 | 1.3382 | 21.4889 | 120.8490 |
| C | BPNN-GA | **0.0273** | 0.1376 | **0.1651** | **0.3272** | 13.7552 |
| | WNN | **0.0273** | **0.1330** | 0.1652 | 0.3274 | **13.3042** |
| | GRNN | 0.0338 | 0.1713 | 0.1839 | 0.4058 | 17.1327 |
| E | BPNN-GA | 0.0054 | 0.0617 | 0.0733 | 0.0645 | 6.1665 |
| | WNN | 0.0055 | 0.0626 | 0.0745 | 0.0665 | 6.2620 |
| | GRNN | **0.0048** | **0.0577** | **0.0696** | **0.0582** | **5.7701** |

**Note:**
Best performers are in bold fonts.

**Table 3 Comparison of statistical significance tests in the prediction results.**

| Hepatitis | Statistic value | BPNN-GA | WNN | GRNN |
|---|---|---|---|---|
| A | R | 0.8992 | 0.9686 | 0.9129 |
| | p-value | 0.00006969 | 0.00000023 | 0.00003383 |
| B | R | 0.9916 | 0.9575 | 0.8030 |
| | p-value | 0.00000000 | 0.00000102 | 0.00166221 |
| C | R | 0.9991 | 0.9141 | 0.6903 |
| | p-value | 0.00000000 | 0.00003198 | 0.01295323 |
| E | R | 0.9409 | 0.9835 | 0.9847 |
| | p-value | 0.00000510 | 0.00000001 | 0.00000001 |

**Note:**
R is correlation coefficient.

The same in regard to the GRNN and WNN method; they had the best correlation when they were used to predict hepatitis E and A, respectively.

$P < 0.01$ are for all models from Table 3 which reveals that the difference is statistically significant between the predictive value and the original data.

## CONCLUSION

This research compared and evaluated the prediction of hepatitis by the BPNN-GA, GRNN, and WNN methods. The prediction results will be affected by the data features. When the small data set has seasonal fluctuation, the prediction result by BPNN-GA will be better than the two other methods. The WNN method is suitable for predicting the large data set that has seasonal fluctuation and the same for the GRNN method when the data increases steadily. The results of all three methods show that the greater the disease fluctuations, the worse the prediction results.

The forecasting efficacies of three models are compared based on performances. GRNN is learns faster and converges to the optimal regression surface. Capturing the dynamic

behavior of hepatitis incidence. Although the BPNN is easy to fall into the local optimum and has highly non-linear weight update and slow coverage rate, the accuracy of forecasting could be improved by optimizing the initial weights and thresholds. The advantage of the BPNN is that it is suited for prediction the small data set has seasonal fluctuation. Compared to BPNN-GA and GRNN, WNN has the best performance when it is used to predict large data set with seasonal fluctuation.

This study can be extended in different directions. First, only hepatitis incidence is predicted in the paper. In order to ascertain performance of three models and possible factors that will impact on the model performance in practice, more infectious diseases should be considered. Finally, we limited the analysis to only three ANN methods, and in future studies more methods could be tested to predict incidence of important diseases, including hepatitis.

### Funding

This work was supported by the project of basic ability promotion for young teachers of Guangxi education department (KY2016YB093). The funders had no role in study design, data collection and analysis, decision to publish, or preparation of the manuscript.

### Grant Disclosures

The following grant information was disclosed by the authors:
Guangxi education department: KY2016YB093.

### Competing Interests

The authors declare that they have no competing interests.

### Author Contributions

- Ruijing Gan performed the experiments, analyzed the data, wrote the paper, prepared figures and/or tables, reviewed drafts of the paper.
- Ni Chen performed the experiments, contributed reagents/materials/analysis tools.
- Daizheng Huang conceived and designed the experiments, performed the experiments, analyzed the data, contributed reagents/materials/analysis tools, wrote the paper, prepared figures and/or tables.

### Data Deposition

The raw data has been supplied as Supplemental Dataset Files.

### Supplemental Information

Supplemental information for this article can be found online at http://dx.doi.org/10.7717/peerj.2684#supplemental-information.

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
