# Peer review of "Comparisons of forecasting for hepatitis in Guangxi Province, China by using three neural networks models"

_PeerJ, doi:10.7717/peerj.2684_

## Round 0.1 · original submission · Major Revisions

Your submission "Comparisons of forecasting for hepatitis in Guangxi province, China by using three neural networks models" is currently not of sufficient quality to be published, in the actual form. Moreover statistical significance tests should be provided by the authors for the performed comparisons. The authors have to improve the study and the manuscript, by following the indications of the Reviewers comments.

·

Basic reporting

Gan et al. propose the application of three neural network classifiers in order to predict the incidence of Hepatitis A,B,C,D, and E in the province of Guangxi (China).
The work is interesting and the prediction results are promising. Although the approach seems to be sound, my main concern of the work is the validation of the results. Are the results and the performed comparisons statistically significant? The authors should provide a statistical significance test. Additionally, the sampling strategy for the training of the neural networks is not well explained. How many folds were selected for cross validations?

Furthermore, the abstract has to be rewritten by focusing on the main objective of the work (i.e., the prediction of the incidence of hepatitis) and by not listing all the evaluation metrics. The results have to be presented in a more clear way better explaining the different metrics used to evaluate the approach. Finally, the conclusions have to be extended by stating what are the main findings of the work and by including the perspectives of it.

Clear, unambiguous, professional English language used throughout.
The English language is ok

Intro & background to show context.
The introduction is good, but some references are missing.

Literature well referenced & relevant.
Yes, but authors should cite other papers that deal with viruses classification, e.g.:
Weitschek, E., Presti, A. L., Drovandi, G., Felici, G., Ciccozzi, M., Ciotti, M., & Bertolazzi, P. (2012). Human polyomaviruses identification by logic mining techniques. Virology journal, 9(1), 1.

Structure conforms to PeerJ standard, discipline norm, or improved for clarity.
Yes

Figures are relevant, high quality, well labelled & described.
The caption of the figures and tables have to extended and improved.

Raw data supplied (See PeerJ policy).
Raw data is supplied in pdf, please provide in csv (comma separated values) spreadsheet.

Experimental design

Original primary research within Scope of the journal.
Yes

Research question well defined, relevant & meaningful.
Can be improved

It is stated how research fills an identified knowledge gap.
No

Rigorous investigation performed to a high technical & ethical standard.
A statistical test is missing.

Methods described with sufficient detail & information to replicate.
Yes.
Minor: on line 113 what is the unit of measure of the time (1000)?; line 117 how many folds were selected for the cross validations

Validity of the findings

Impact and novelty not assessed.
The impact and novelty are low, but the article is interesting.

Negative/inconclusive results accepted.
No negative results

Meaningful replication encouraged where rationale & benefit to literature is clearly stated.
No replication

Data is robust, statistically sound, & controlled.
A statistical significance test is missing

Conclusion well stated, linked to original research question & limited to supporting results.
Conclusions have to be improved and extended.

Speculation is welcome, but should be identified as such.
No speculation

Reviewer 2 ·

Basic reporting

the article is written in good english.
the introduction is sufficient.
The description of the 3 methods is very synthetic and not sufficient to make the paper self contained. this is true, in particular, for the description of the third method.
the structure is compliant with a template.
figure are relevant and properly described and labeled.

Experimental design

the research question is well posed but incomplete.
authors aim at comparing 3 forecasting methods based on several parameters; they do not provide justification for the choice of the 3 methods as opposed to other classical forecasting methods well established in the statistics literature (e.g., autoregressive models such as Arima and Arma). Such methods are quoted in the literature review but not tested, nor results obtained in the paper are compared with those from literature.
there is to little mentioning of tuning parameters, of methods, softwares, and algorithms used for the estimation of the models.
additionally, it is not clear how cross validation is used in the context of time serie and with how many folders.
form the description of the results it is not clear if the 3 models differ in the variables that are used or in the lag, or whether a search on the optimal lag of the autoregressive model is performed.

Validity of the findings

the findings may have some interesting edges; alhtough, the differences in the performances of the tested models/methods are not so marked; in order to establish dominance between them, some statistical considerations (hypothesis testing) should be performed to compare the indicators. Also, authors should test properly tuned autoregressive simples model to make sure the the resort to much more complex and unstable methods such as the ones based on NN are indeed demanded by the complexity of the data.

Additional comments

the quality of the writing is good; few typos and improvable details are present but I will pospone their fixing to an possible second submisson of the paper. In such a revision, make sure that the experiments may be replicated by a third party providing all needed information.

---

## Round 0.2 · Minor Revisions

The manuscript was improved by the authors but further changes need to be done. I suggest to follow the indications and comments of the two Reviewers to improve the manuscript.

·

Basic reporting

The authors did a good job in improving the manuscript, but some additional minor revisions have to be done:

1. Carefully check the manuscript for typos and for the English language style
2. Use the present tense for the abstract
3. In the keywords change “forcasting” to “forecasting”
4. Reference number (1) is in the wrong place, please add a sentence at the end of the introduction, where you state that other studies have been performed with supervised methods for predicting viruses and pathologies and cite reference number 1 and other related works
5. The caption of the figures and tables have to extended and improved.
6. Line 175: insert the unit of measure of time (iterations) in the manuscript
7. Line 184: why did you chose 4 folds for the cross validation sampling strategy?

Clear, unambiguous, professional English language used throughout.
The English language is ok, but some improvements can be done

Intro & background to show context.
The introduction is good, but some references are missing.

Literature well referenced & relevant.
Yes, but authors should cite other papers that deal with viruses classification and
state that other studies have been performed with supervised methods for predicting viruses and pathologies.

Structure conforms to PeerJ standard, discipline norm, or improved for clarity.
Yes

Figures are relevant, high quality, well labelled & described.
The caption of the figures and tables have to extended and improved.

Raw data supplied (See PeerJ policy).
Yes

Experimental design

Original primary research within Scope of the journal.
Yes

Research question well defined, relevant & meaningful.
Yes

It is stated how research fills an identified knowledge gap.
No

Rigorous investigation performed to a high technical & ethical standard.
Yes

Methods described with sufficient detail & information to replicate.
Yes.

Minor: on line 175 what is the unit of measure of the time (1000)?; line 184 why do you chose 4 folds?

Validity of the findings

Impact and novelty not assessed.
The impact and novelty are low, but the article is interesting.

Negative/inconclusive results accepted.
No negative results

Meaningful replication encouraged where rationale & benefit to literature is clearly stated.
No replication

Data is robust, statistically sound, & controlled.
A statistical significance test is missing

Conclusion well stated, linked to original research question & limited to supporting results.
Conclusions have to be improved and extended.

Speculation is welcome, but should be identified as such.
No speculation

Reviewer 2 ·

Basic reporting

suggested revisions have been taken into account almost completely. The paper is compliant with the main requirements

Experimental design

the experimental design is satisfactory

Validity of the findings

The findings are interesting and properly organized.

Additional comments

Please consider the attached PDF where several minor revisions are suggested, as revision of the word document. I suggest an additional final revision of the english by a native speaker.

Annotated reviews are not available for download in order to protect the identity of reviewers who chose to remain anonymous.

---

## Round 0.3 · Minor Revisions

I think that this study is almost ready for publication, the authors only have to adjust some remaining sentences, references and words to improve the text, as suggested by the reviewer.

·

Basic reporting

The revised manuscript has been substantially improved and hence I suggest to accept it for publication after following minor revisions have been performed.

Minor revisions:
Line 34: missing space before the "(" and cite other studies (e.g., E. Weitschek, F. Cunial, G. Felici: LAF: Logic Alignment Free and its application to bacterial genomes classification. Biodata Mining, 8(1):39, 2015.
D. Polychronopoulos, E. Weitschek, S. Dimitrieva, P. Bucher, G. Felici, Y. Almirantis: Classification of selectively constrained DNA elements using feature vectors and rule-based classifiers. Elsevier Genomics, 104(2):79-86, 2014.)
Line 53: delete the word "of"
Line 81: change "Figure 2 The prediction algorithm of BPNN optimized by GA flow chart." to "Figure 2 Flow chart of the BPNN prediction algorithm optimized by GA ."
Line 125: change "Figure 4 The prediction algorithm of WNN flow chart." to "Figure 4 Flow chart of the WNN prediction algorithm."
Line 178: change "Table 1 Parameters of GA Using to Optimize BPNN" to "Table 1 Parameters of the GA used to optimize the BPNN."
Line 250 change "Table 2 Comparison of Evaluation Indexes of Prediction Results" to "Table 2 Comparison of the evaluation indexes in the prediction results"
Line 272 change "Table 3 Comparison of Statistical Significance Tests of Prediction Results" to "Table 3 Comparison of Statistical Significance Tests in the prediction results."
Line 292: change "suit" to "suited"
Line 296: change "could" to "can" and "was to" to "is"
Line 297: change "predict" to "predicted"
Line 299: change "Second," to "Finally,"

Clear, unambiguous, professional English language used throughout.
The English language is ok.

Intro & background to show context.
The introduction is good.

Literature well referenced & relevant.
Yes.

Structure conforms to PeerJ standard, discipline norm, or improved for clarity.
Yes

Figures are relevant, high quality, well labelled & described.
Yes

Raw data supplied (See PeerJ policy).
Yes

Experimental design

Original primary research within Scope of the journal.
Yes

Research question well defined, relevant & meaningful.
Yes

It is stated how research fills an identified knowledge gap.
Yes

Rigorous investigation performed to a high technical & ethical standard.
Yes

Methods described with sufficient detail & information to replicate.
Yes

Validity of the findings

Impact and novelty not assessed.
The impact and novelty are low, but the article is interesting.

Negative/inconclusive results accepted.
No negative results

Meaningful replication encouraged where rationale & benefit to literature is clearly stated.
No replication

Data is robust, statistically sound, & controlled.
Yes

Conclusion well stated, linked to original research question & limited to supporting results.
Yes

Speculation is welcome, but should be identified as such.
No speculation

---

## Round 0.4 · accepted · Accept

This study "Comparisons of forecasting for hepatitis in Guangxi province, China by using three neural networks models" is now ready for publication, the authors have improved the text according to the reviewer's and editor comments.